# Effect of Extrusion Temperature on the Microstructure and Mechanical Properties of SiCnw/2024Al Composite

**DOI:** 10.3390/ma12172769

**Published:** 2019-08-28

**Authors:** Shanliang Dong, Bin Zhang, Yuli Zhan, Xin Liu, Ling Xin, Wenshu Yang, Gaohui Wu

**Affiliations:** 1Harbin Institute of Technology, Harbin 150001, China; 2Beijing Institute of Control Engineering, Beijing 100080, China

**Keywords:** metal matrix composites, Al matrix composite, SiC nanowires, extrusion temperature, alignment, strengthening

## Abstract

In the present research work, the effect of extrusion temperature from 480 to 560 °C on the microstructure and mechanical behavior of the SiCnw/2024Al composite (15 vol.%) has been explored. It has been found that extrusion at higher temperature (above 520 °C) was beneficial for the densification of the composite, while the residual average length and alignment of the SiC nanowires were also increased with the extrusion temperature. Moreover, higher extrusion temperature was helpful for the mechanical strength of the SiCnw/2024Al composite, and the peak-aged SiCnw/2024Al composite extruded at 560 °C revealed the highest strength (709.4 MPa) and elastic modulus (109.8 GPa).

## 1. Introduction

Recently, composites materials reinforced with nano-phases, such as ceramic nano-particles [1,2], carbonous nano-phase (CNT) [3,4] and graphene [5,6] have been widely explored. Besides the above phases, SiC nanowires are also very promising to be used as reinforcement materials [7,8]. It has been reported that the SiC nanowires could enhance the strength and toughness of polymer [9] and ceramic matrix composites [10]. 

Utilization of the SiC nanowires to reinforce metal matrix composites has also been reported. Pozuelo et al. [11] found that the SiC nanowires could be well bonded with Mg matrix and suggested that the enhanced interlocking effect between SiC nanowires and Mg matrix would be helpful for the mechanical properties. Lee et al. [12] reported that the SiC nanowires reacted with tungsten matrix to form rod-type W_5_Si_3_ phases, which improved the flexural strength and ablation resistance. Kang et al. [13] also concluded that the SiC nanowires would react with W matrix to form W_2_C and W_5_Si_3_, which improved the compressive strength (from 0.9 to 2.29 GPa) and strain simultaneously. Jintakosol et al. [14] found that the SiC nanowires could increase the wear resistance of the Al matrix composites significantly. Recently, our research group reported that the microstructure and mechanical behavior of SiC nanowires reinforced pure Al and 6061Al matrix composites [15,16,17,18,19,20]. Compared with the carbonaceous materials (CNT, graphene), which easily react with Al matrix to form detrimental Al_4_C_3_ phase [3,21,22], SiC nanowires-Al interface demonstrated good interfacial bonding performance without [18,20] or with minimal interfacial products [19]. Moreover, it has been found that 15 vol.% SiCnw/6061Al composite showed high bending strength (over 1000 MPa) and comparable plasticity as that of Al matrix [15].

Hot-extrusion treatment has been widely used as an important preparation and strengthening method [23,24,25]. In our previous work, it has been reported that the mechanical properties of the SiCnw/6061Al composites could be enhanced by the extrusion treatment regardless of SiC nanowires content [26]. Usually, the extrusion effect was affected by the parameters (extrusion temperature and ratio) and the characters of the composites (aspect ratio and content of the reinforcement). Luan et al. [27] found that the extrusion temperature (445 to 570 °C for Al_2_O_3_p/2024Al and 445 to 570 °C for Al_2_O_3_p/6061Al) showed slight effect on the mechanical properties of Al_2_O_3_p/Al composites. However, Tavighi et al. [28] reported that the tensile strength of the 16 wt.% Al_4_Srp/Al composites was decreased with the extrusion temperature (390 to 500 °C) while the ductility reached maximum at 420 °C. Contrarily, Feng et al. [29] reported that the strength of pure Al matrix composites reinforced with 22 vol.% Al_18_B_4_O_33_ whiskers and 3 vol.% WO_3_ particles was initially increased and then decreased with the extrusion temperature (440 to 560 °C). Feng et al. [29] also observed that the length of Al_18_B_4_O_33_ whiskers in the composites was increased with the temperature. Meanwhile, Hong et al. [30] revealed that the strength of 2024Al composites reinforced with 20 vol% SiC whiskers were increased linearly with the extrusion temperature (470 to 530 °C) due to the improved alignment of the SiC whiskers. However, the effect of the extrusion temperature on the SiCnw/Al composite, with high strength and high aspect ratio reinforcement (more than 50), has not been explored yet.

In the present research work, 15 vol.% SiCnw/2024Al composite has been hot-extruded in the temperature range from 480 to 560 °C, and the effect of the extrusion temperature on the microstructure and mechanical behavior of the SiCnw/2024Al composite has been discussed. 

## 2. Experimental Design, Materials and Methods

### 2.1. Materials and Composite Fabrication Process

The raw SiC nanowire used in the present research was provided by Changsha Sinet Advanced Materials Co., Ltd. Changsha, China, while the 2024Al alloy was supplied by Northeast Light Alloy Co., Ltd. Harbin, China. The chemical composition (wt.%) of the 2024Al alloy was 4.23% Cu, 1.58% Mg, 0.75% Mn, 0.25% Zn, 0.10% Cr and Al balance. The fabrication process of SiCnw/2024Al composite was similar to that of the SiCnw/6061Al composite [15,18], as shown in Figure 1. The preheating temperatures for the preform and graphite indenter were 520 and 760 °C, and the melting temperature of the 2024Al alloy was 840 °C, respectively. Afterwards, the SiCnw/2024Al billets were hot extruded at 480, 500, 520 and 560 °C, respectively. The extrusion ratio and speed were 11:1 and 1 mm/s respectively. Before mechanical test, the SiCnw/2024Al specimens were treated by peak-aging (495 °C/1 h solid solution, water quenched and then 160 °C/8h aging) or annealing treatment (340 °C/1 h).

### 2.2. Microstructure Characterization

Microstructure of SiCnw/2024Al specimens were observed by optical microscope (Axiovert 40 MAT, Carl Zeiss, Jena, Germany), scanning electron microscope (SEM) (FEI Sirion Quanta 200, Philips, Amsterdam, Netherlands) and transmission electron microscope (TEM) (JEM-2010F, JEOL, Tokyo, Japan). Energy Dispersive X-Ray (EDX) Spectroscopy equipped in SEM has been applied to detect the distribution of Al, Si, and C elements, which was further used to reveal the distribution of SiC nanowires. The density of all samples (10 mm × 10 mm × 2 mm) was measured using Archimedes principle, and four samples have been tested to improve the statistical significance of the results. The relative density of the composites before and after extrusion was calculated by comparison of the measured true density and the “theoretical” density of the samples extruded at 560 °C. X-ray diffraction (XRD) was performed on Rigaku D/max-rB diffractometer with Cu-Kα radiation (0.15418 nm) between 15° and 90°. 

### 2.3. Mechanical Properties Testing

Elastic modulus was tested by impulse excitation of vibration on samples with dimensions of 3 mm × 4 mm × 36mm according to ASTM E 1876 (EMT-01, Zhuosheng Instrument Co., Ltd. Luoyang, China). Parallel to the extrusion direction, tensile properties were measured on universal electrical tensile testing machine (Instron 5569, Instron Co., Norwood, MA, USA) with a speed of 0.5 mm/min, and eventually the SEM fracture surface was observed by Quanta 200. The tensile samples (2 mm in thickness) were prepared from the composites, as shown in Figure 2. Six samples extruded at different temperature were tested for each set of composite samples to improve the statistical significance of the results.

## 3. Results and Discussion

### 3.1. Effect of Extrusion Temperature on the Microstructure of SiCnw/2024Al Composite

Due to the large difference between the size of the SiC nanowires (about sub-micron) and pores (about 20 μm), it is difficult to show them both clearly in the same image. In order to show the effect of the extrusion treatment on the distribution of pores in the composites, optical morphologies of SiCnw/2024Al composite extruded at different temperature have been shown in Figure 3. Several pores, which have been marked out by the dashed circles, were found in the as-casted SiCnw/2024Al composite (Figure 3a). It was mainly due to the relatively poor wettability between SiC and Al matrix [31]. Zeng et al [32] reported that the increased sintering temperature was beneficial for the densification of Si_3_N_4_w/Al composites. Meanwhile, Zeng et al. [33] found innovatively that the Ag surface modification of the Si_3_N_4_ whiskers could improve the relative density of the Si_3_N_4_w/Al composite, and Zeng et al. [33] attributed the effect to the lower eutectic point of Al-Ag interface, improved wetting behavior, and broken down of the Al_2_O_3_. These interesting results would be referable for further improving of the densification result. After extrusion treatment, the porosity of the SiCnw/2024Al composite has been significantly decreased (Figure 3b,c). It has been widely reported that one of the main advantages of the hot-extrusion treatment is the densification [23,25,29]. Moreover, the porosity of the SiCnw/2024Al composite has been almost eliminated after extrusion treatment at 520 °C and 560 °C (Figure 3d,e), indicating more effective densification process at higher temperatures due to the good deformability of the Al matrix [30]. 

After extrusion, the measured density of the composites was initially increased with and eventually stabilized above 520 °C with the increase of the extrusion temperature, indicating that the composites have reached full densification after extrusion at 560 °C. Therefore, the measured true density of the composites in various states were divided to the “theoretical” density of the samples extruded at 560 °C, and then the relative density has been obtained, as shown in Figure 4. It is clear that the composites have been densified after the extrusion, and the densification effect was also improved with the extrusion temperature, which agreed well with optical microstructure observation (Figure 2).

In order to reveal the distribution of the SiC nanowires in the composites, EDX mapping has been performed, and the distribution of the Al, Si and C elements have been shown in Figure 5. Due to the accuracy of the EDX, the distribution of C element was not very reliable. However, the distribution of Si element was very clear, as shown in Figure 5c. Due to the low content of Si element in 2024Al matrix, the distribution of Si, mainly in morphology of rod-like structures, could be considered to be equivalent to distribution of SiC nanowires. This is also in agreement with our previous research work [15,17] and literature results [11,14]. 

Figure 6 shows the effect of extrusion treatment on the SEM alignments of the SiC nanowires in the composite. In order to observe the SiC nanowires’ distribution clearer, the polished samples were etched by 10% NaOH solution for 120 s. Since the results were quite similar regardless of the extrusion temperature, only the SiC nanowires alignments extruded at 560 °C (Figure 5b,c) were exhibited. Before extrusion treatment, SiC nanowires towards different directions were observed (Figure 6a), indicating their random distribution. However, after extrusion treatment, the well aligned ends of the SiC nanowires were observed to be perpendicular to the extrusion direction (Figure 6b), while significant orientation of the SiC nanowires along extrusion axis has been found to be parallel to the extrusion direction (Figure 6c), indicating the well alignment of the SiC nanowires along the extrusion direction. Attributed to the rheology shear stress of the Al matrix, the reinforcements are prone to being re-distributed along the extrusion direction. Moreover, it has been widely reported that the extrusion treatment would cause the breakage of high aspect ratio reinforcements (such as whiskers [29,34]) due to the mismatch of the deformation ability. In order to measure the average length of the SiC nanowires after extrusion, the microstructure of SiCnw/2024Al composite extruded at 560 °C after being etched by 10% NaOH solution for 480 s is shown in Figure 7a, and eventually the length of the SiC nanowires was counted by the Nano Measurer software. After statistical analysis, the normal distribution of the SiC nanowires’ length could be given by the software, as shown in Figure 7b. The effect of extrusion temperature on the average length of the SiC nanowires has been shown in Figure 8. Regardless of the extrusion temperature, it is clear that the average length of the SiC nanowires after the extrusion treatment was about half of that in the as-casted composite (Figure 8), indicating severe breakage of the SiC nanowires during the extrusion treatment. During the hot extrusion process, the Al matrix could be deformed plastically while the SiC nanowires were considered as non-deformable phases, which would be rotated due to the rheological stress of the Al matrix, and then the breakage of the SiC nanowires could occur. For the reinforcement with large size (such as Al_4_Sr phase with about 30 μm in thickness [28]), the areas between the broken phases could not be re-filled fully by the Al matrix, and they would therefore act as crack sources and thus decrease the mechanical properties of the composites. However, for the reinforcement with small size (diameter less than a few microns) [27,29], the areas between the broken reinforcement could be healed by the Al matrix. Thus, the extruded composites would express rather high relative density and high mechanical performance. However, it is worth noting that the average length of the SiC nanowires was increased with the extrusion temperature (Figure 8). This agreed well with the results reported by Feng et al. [29] and Hong et al. [30]. It was mainly due to the decreased deformation resistance of the Al matrix on the rotation of the SiC nanowires at higher extrusion temperature [30].

The XRD patterns of the extruded SiCnw/2024Al composite, whose analyzed surface was perpendicular and parallel to the extrusion direction, have been shown in Figure 9 and Figure 10, respectively. Regardless of the testing direction, only diffraction peaks of SiC and Al were detected (Figure 9a and Figure 10a). The corresponding diffraction peaks of the (111) and (220) planes of the SiC nanowires have been shown in Figure 9b/Figure 10b and Figure 9c/Figure 10c, respectively. The peak value ratio (PVR) of the XRD patterns of the (111) and (220) planes of the SiC nanowires were calculated as:(1)$$\mathrm{PVR}=\frac{I_{\left(111\right)}}{I_{\left(220\right)}}$$
where *I* is the peak value of the XRD pattern, the subscript (111) and (220) represent the (111) and (220) planes, respectively. 

The PVR of the SiC nanowires in the SiCnw/2024Al composite in perpendicular and parallel to the extrusion direction has been shown in Figure 11. In the perpendicular direction, the PVR of the SiC nanowires was increased with the extrusion temperature, indicating the increased diffraction intensity of the (111) planes. However, in the parallel extrusion, the PVR of the SiC nanowires was decreased with the extrusion temperature, indicating the increased diffraction intensity of the (220) planes [111] axes is the axial direction of the SiC nanowires, while the (111) planes are the stacking planes of the 3C-SiC [35,36]. The increased PVR in the perpendicular direction and the decreased PVR in the parallel direction indicated the increased alignment degree of the SiC nanowires along the extrusion direction. Therefore, alignment degree of the SiC nanowires was increased with the extrusion temperature, which is consistent with the results reported by Feng et al. [29] and Hong et al. [30]. 

The microstructure of extruded SiCnw/2024Al composite after annealing has been further observed by TEM, as shown in Figure 12. The SiC nanowires, as marked out by arrow in Figure 12a, were mainly perpendicular to the observation direction. The selected area electron diffraction (SAED) patterns of the dark phase (as marked out by yellow dashed circle) have been shown in Figure 12b. It is clear that two set patterns could be found, while the central spots and the second hexagonal spots of the SAED of the SiC nanowires were very bright, and the first hexagonal spots were relatively dark (Figure 12b). In our previous research work [37], the cross-section and axial microstructure of the SiC nanowires embedding into Al matrix have been investigated, and it has been found that the SiC nanowires were composed of a large number of small fragments that are formed by hybrid 3C-SiC and 2H-SiC structures. Therefore, their SAED patterns were composed of two sets of diffraction patterns, corresponding to 2H-SiC and 3C-SiC, respectively, which agreed well with the present observation. Moreover, high magnification image indicated that the presence of the SiC nanowires was mainly located at the boundary of Al grains, as shown in Figure 12c. Usually, for SiCnw/Al composites prepared by pressure infiltration method, the SiC nanowires were mainly located within the Al grains due to the solidification of the matrix [15,20]. It is suggested that the “non-deformable” SiC nanowires inhabited the movement and deformation of surrounding Al matrix, leading to the formation of the grain boundaries [26,29]. The refinement of the Al matrix grain would also be beneficial for the mechanical properties. However, a few non-directional SiC nanowires could have also been observed in the composites, as shown in Figure 12d. 

### 3.2. Effect of Extrusion Temperature on the Mechanical Behavior of SiCnw/2024Al Composite

The representative tensile curves of annealed and peak-aged SiCnw/2024Al composite extruded at different temperature have been shown in Figure 13. The annealed SiCnw/2024Al composite showed very low plastic deformation while the peak-aged SiCnw/2024Al composite showed the typical characters of the brittle fracture. The detailed comparison of the yield and tensile strength has been shown in Figure 14. Since the peak-aged SiCnw/2024Al composite was fractured before yield deformation, its yield strength has not been obtained. Regardless of the heat treatment, the yield and tensile strengths of the SiCnw/2024Al composite were increased with the extrusion temperature. Moreover, the peak-aged SiCnw/2024Al composite showed higher tensile strength, which should be attributed to the strengthening effect of the precipitates. Fracture surface of SiCnw/2024Al composite extruded at 560 °C has been shown in Figure 15. Many dimples and tearing ridges were observed in the fracture surface (Figure 15a), indicating severe deformation of Al matrix. Moreover, several broken and pull-out SiC nanowires, which have been pointed out by yellow arrows in Figure 15b, were also observed. Meanwhile, the elastic moduli of the SiCnw/2024Al composite before and after the extrusion treatment were 101.3 and 109.8 GPa, respectively. However, for the composites after extrusion, the extrusion temperature showed minimal effect on the elastic modulus of the SiCnw/2024Al composite. The increased elastic modulus was mainly due to the improved relative density [26,38]. Regardless of the extrusion temperature, the SiCnw/2024Al composite after extrusion treatment could be considered to be fully densified (Figure 4), leading to the slight variation of the elastic modulus at different extrusion temperature. The peak-aged SiCnw/2024Al composite showed high strength (709.4 MPa) and elastic modulus (109.8 GPa), which are very attractive features for aerospace application. 

The strength of discontinuous phases reinforce metal matrix composites could be well explained by the modified shear-lag model after taking into account the tensile transfer of the load from the matrix to the discontinuous reinforcement [23,39]. Moreover, it has been suggested that the strengthening behavior of the SiCnw/Al composites could be well described by the modified shear-lag model which considers the effect of the aspect ratio of reinforcements (*l*/2*r*), reinforcement-matrix interfacial bonding properties *k*, the surface-to-volume ratio of the reinforcement (the geometry factor, *g*) and the average angle between the loading direction and the reinforcement axis (the alignment factor, *ψ*) [20,26,40]. If the length of the SiC nanowires (*l*) is longer than the critical length (*l_c_*), then the strengthening behavior of the SiCnw/Al composites could be written as:(2)$$\sigma_{c}=\sigma_{m}+ckg\psi \frac{l}{2r}(2-\frac{l_{c}}{l})\tau_{\max}V_{r}$$
where *σ_c_* and *σ_m_* are the strengths of the composite and matrix, *V_r_* is the volume amount of reinforcement, *c* is an empirical constant for the correction, *r* is radius of the reinforcements, *τ*_max_ is maximum shear stress, respectively. 

Due to comprehensive effect of the helpful aspects (decreased porosity, increased alignment) and detrimental aspect (decreased average length), the strength of the SiCnw/2024Al composite has been improved after the extrusion treatment. For the well densified SiCnw/2024Al composite after extrusion, the *σ_m_*, *r*, *k*, *g*, *l_c_*, *V_r_* and *τ*_max_ could be considered as the same parameters. Moreover, the constant *c* might be solely dependent on the bonding properties of reinforcement elements [40]. Therefore, the strength of the composites after extrusion was mainly affected by the alignment factor *ψ* and the length of the SiC nanowires *l*. Since the alignment (Figure 11) and the average length (Figure 7) of the SiC nanowires were increased, the strength of the SiCnw/2024Al composite was increased with the extrusion temperature. It should be noted that the peak-aged SiCnw/2024Al composite was failure before yield deformation (0.2%) and the yield strength of the SiCnw/2024Al composite has the potential to be more than 700 MPa, optimistically, if the composites demonstrated certain plasticity. Therefore, the main challenge for SiCnw/Al composites to achieve high strength is to improve the ductility. 

## 4. Conclusions

In the present work, the effect of extrusion temperature from 480 to 560 °C on the microstructure and mechanical behavior of the 15 vol.% SiCnw/2024Al composite prepared by pressure infiltration methods have been investigated, and following conclusions have been obtained:After extrusion treatment, the porosity of the SiCnw/2024Al composite has been significantly decreased, and the composites have reached full densification after extrusion at 560 °C. Moreover, the average length of the residual SiC nanowires was also increased with the extrusion temperature.The distribution of the SiC nanowires has been changed from random to alignment along the extrusion direction, while the alignment degree of the SiC nanowires was increased with the extrusion temperature according to XRD analysis. TEM observation indicated that the presence of the SiC nanowires was mainly located at the boundary of Al grains due to their “non-deformable” character.Regardless of the heat treatment, the yield and tensile strengths of the SiCnw/2024Al composite were increased with the extrusion temperature, and many dimples and tearing ridges were observed in the fracture surface. The peak-aged SiCnw/2024Al composite extruded at 560 °C revealed highest strength (709.4 MPa) and elastic modulus (109.8 GPa).Based on the modified shear-lag mode, the effect of the extrusion temperature on the mechanical properties of the SiCnw/2024Al composite has been discussed, and the improved strength was mainly due to the better alignment and longer average length of the SiC nanowires at higher extrusion temperature.

## Figures and Tables

**Figure 1 materials-12-02769-f001:**
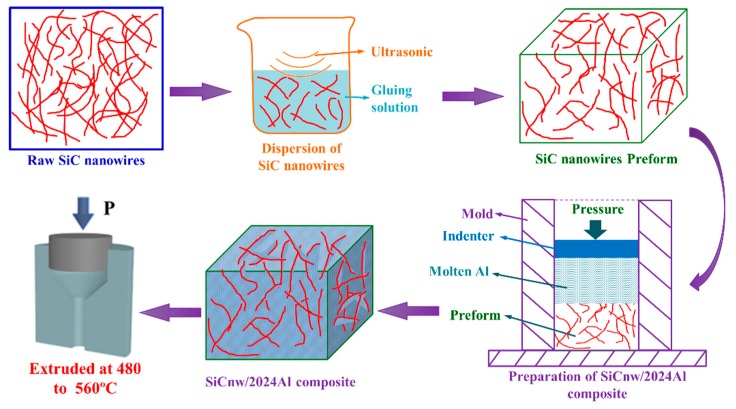
The schematic diagram of preparation and extrusion process of SiCnw/2024Al composite.

**Figure 2 materials-12-02769-f002:**
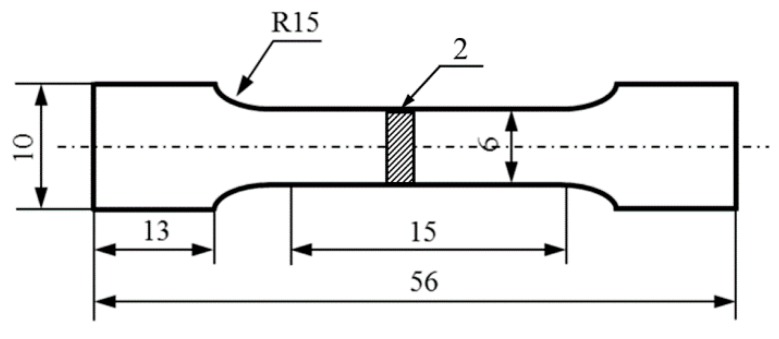
Dimensions of tensile test samples (2 mm in thickness).

**Figure 3 materials-12-02769-f003:**
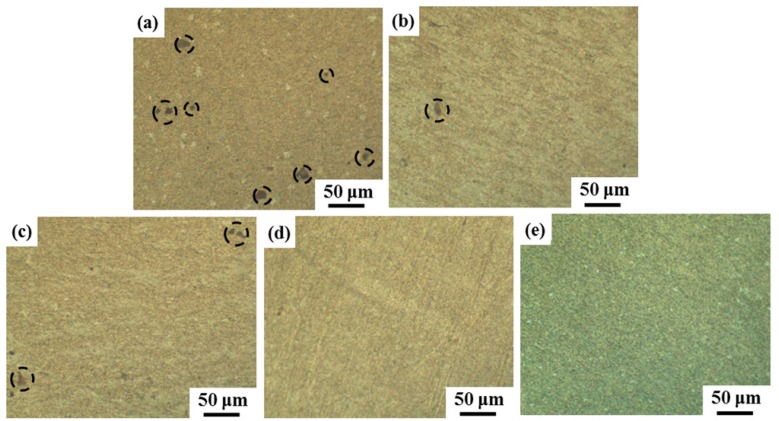
Representative optical morphologies of SiCnw/2024Al composite (**a**) before extrusion and extruded at (**b**) 480 °C, (**c**) 500 °C, (**d**) 520 °C and (**e**) 560 °C.

**Figure 4 materials-12-02769-f004:**
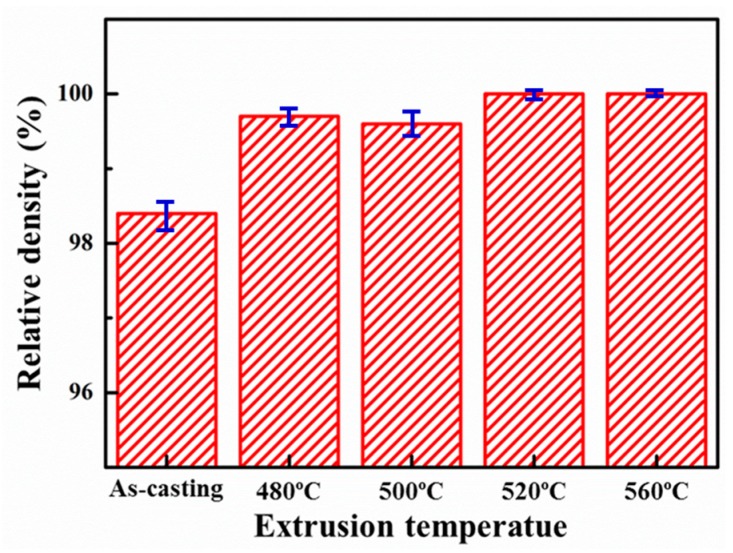
Effect of extrusion temperature on the relative density of the SiCnw/2024Al composite.

**Figure 5 materials-12-02769-f005:**
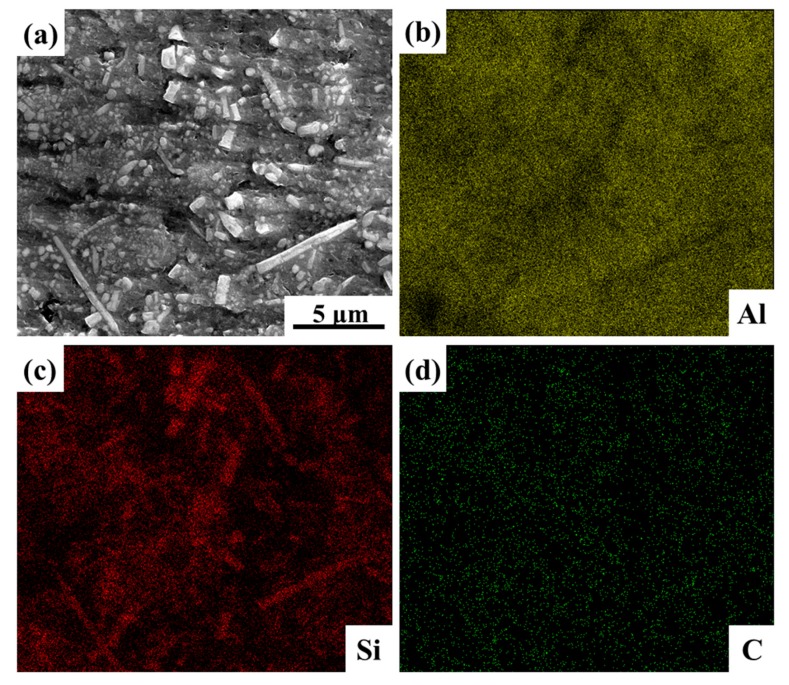
Energy Dispersive X-Ray (EDX) mapping of the polished SiCnw/2024Al composite. (**a**) SEM image and corresponding distribution of (**b**) Al, (**c**) Si and (**d**) C, respectively.

**Figure 6 materials-12-02769-f006:**
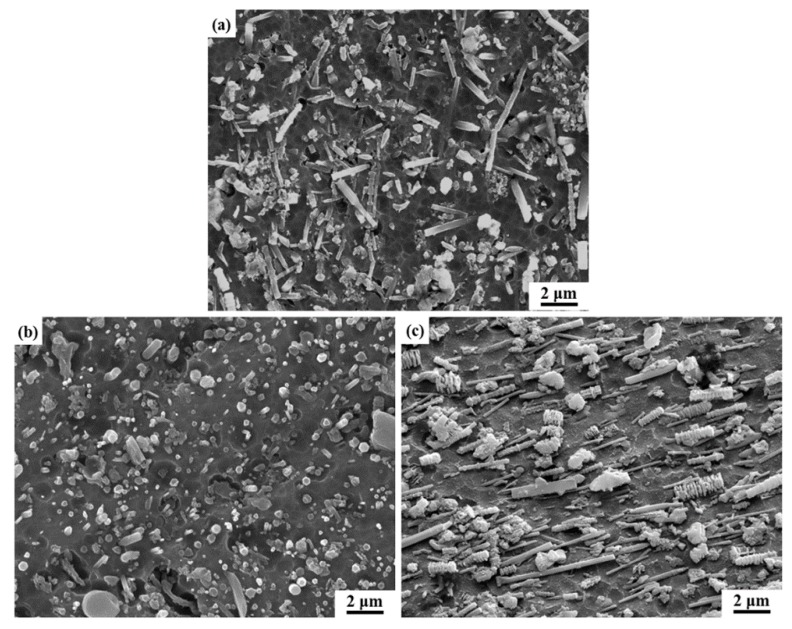
The representative SEM microstructure of the SiCnw/2024Al composite (**a**) before and after 560 °C extrusion observed in the direction (**b**) perpendicular and (**c**) parallel to the extrusion direction.

**Figure 7 materials-12-02769-f007:**
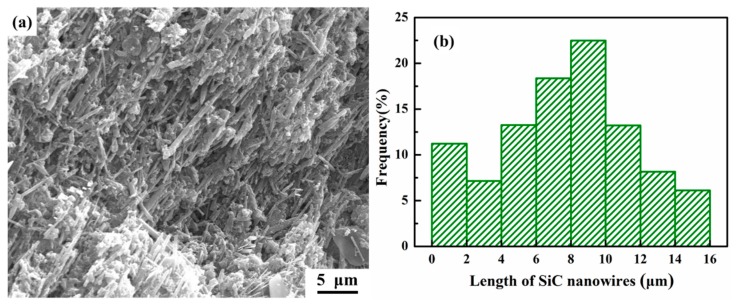
Microstructure of SiCnw/2024Al composite extruded at 560 °C after being etched by 10% NaOH solution for 480 s and corresponding statistical results of the SiC nanowires’ length. (**a**) Microstructure, (**b**) Statistical results.

**Figure 8 materials-12-02769-f008:**
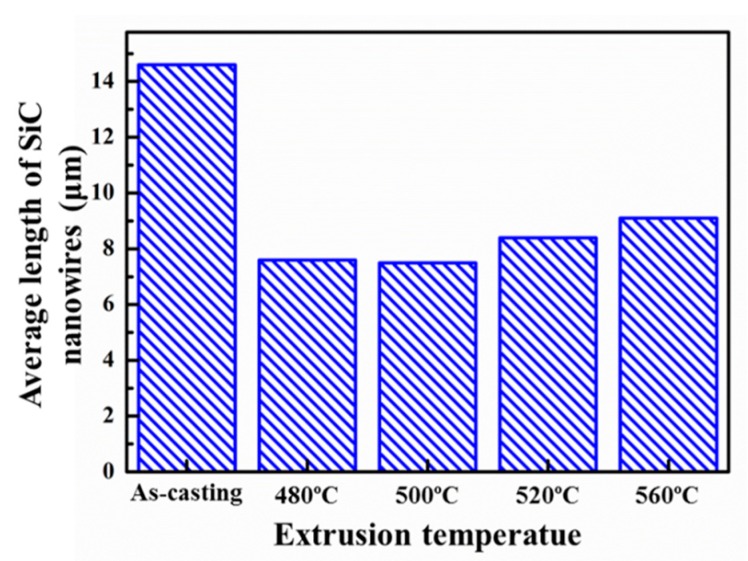
The effect of extrusion temperature on the average length of the SiC nanowires in the SiCnw/2024Al composite.

**Figure 9 materials-12-02769-f009:**
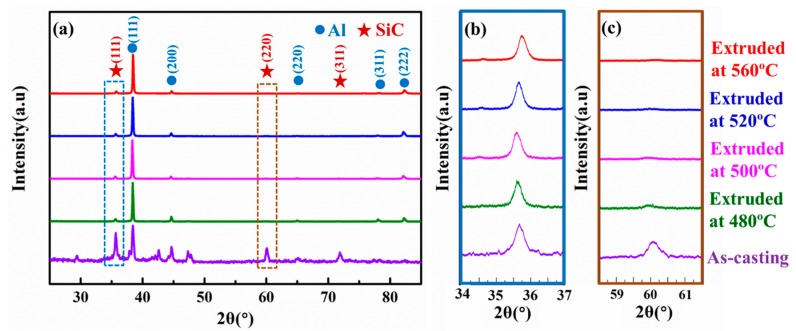
The XRD patterns of the SiCnw/2024Al composite before and after extrusion treatment on the surface perpendicular to the extrusion direction. (**a**) Overall patterns and the high magnification of the (**b**) (111) planes and (**c**) (220) of the SiC phase.

**Figure 10 materials-12-02769-f010:**
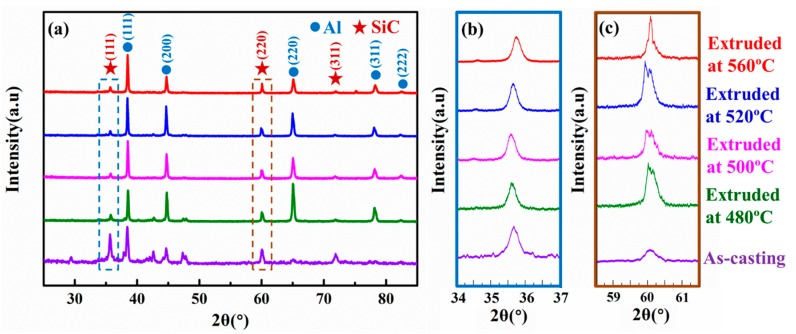
The XRD patterns of the SiCnw/2024Al composite before and after extrusion treatment on the surface parallel to the extrusion direction. (**a**) Overall patterns and the high magnification of the (**b**) (111) planes and (**c**) (220) of the SiC phase.

**Figure 11 materials-12-02769-f011:**
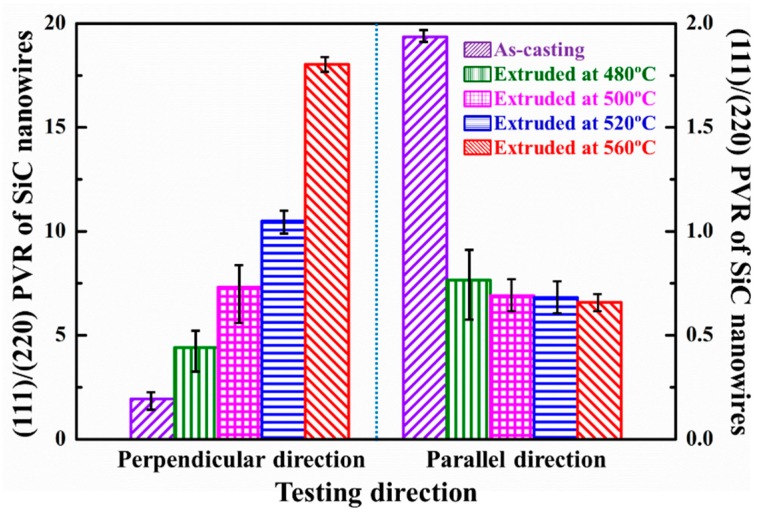
The (111)/(220) peak value ratio (PVR) of the SiC nanowires in the SiCnw/2024Al composite in perpendicular and parallel to the extrusion direction.

**Figure 12 materials-12-02769-f012:**
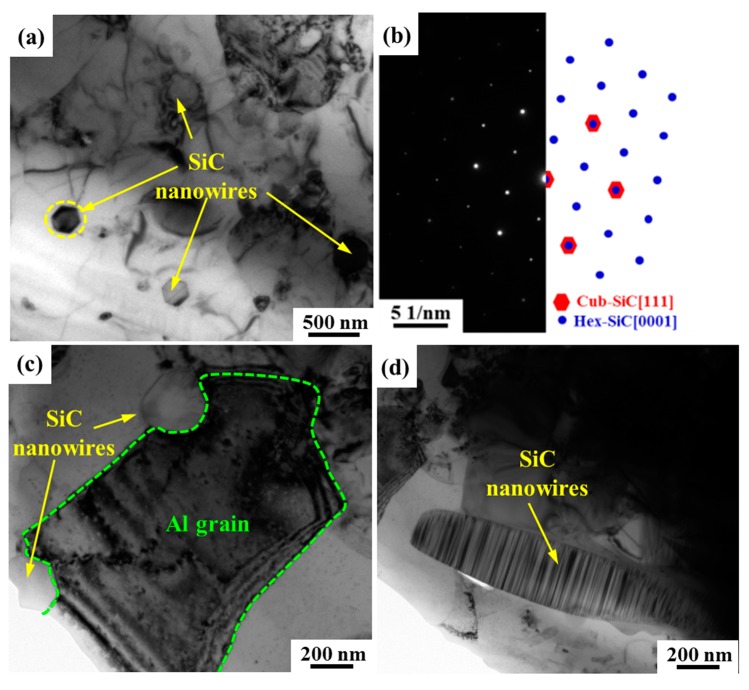
The TEM microstructure of extruded SiCnw/2024Al composite after annealing. (**a**) Low magnification image, (**b**) selected area electron diffraction (SAED) patterns of the SiC nanowire in the dashed circle and (**c**) SiC nanowires located at the boundaries of Al grains, (**d**) few non-directional SiC nanowires.

**Figure 13 materials-12-02769-f013:**
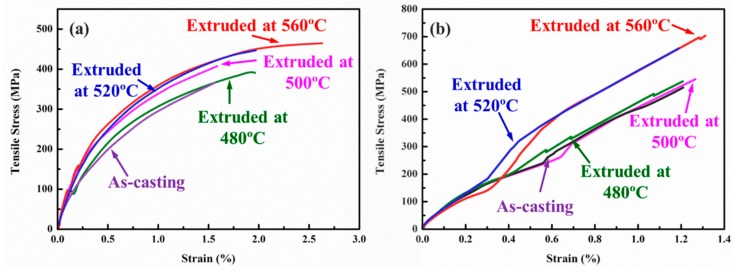
The representative tensile curves of (**a**) annealed and (**b**) peak-aged SiCnw/2024Al composite extruded at different temperature.

**Figure 14 materials-12-02769-f014:**
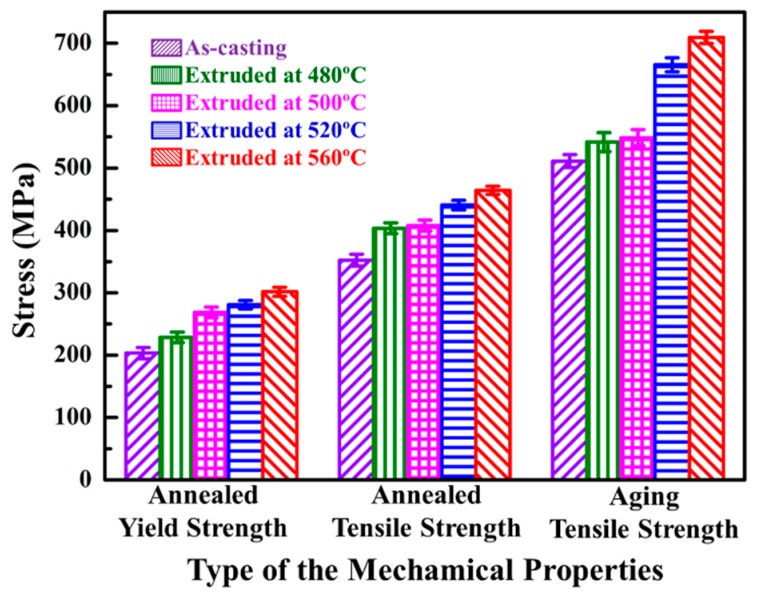
The detailed comparison of the yield and tensile strength of the SiCnw/2024Al composite.

**Figure 15 materials-12-02769-f015:**
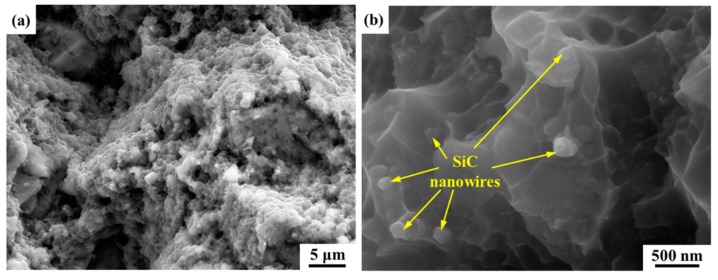
Fracture surface of SiCnw/2024Al composite extruded at 560 °C.

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
