# Peer review of "Effect of Extrusion Temperature on the Microstructure and Mechanical Properties of SiCnw/2024Al Composite"

_materials, 2019, doi:10.3390/ma12172769_

Round 1
Reviewer 1 Report
This paper is not recommended for publication for the following reasons:
1. English is very poor, should be edited professionally
2. Optical microscopy images shown do not provide any morphology insights
3. SEM images (Fig 3, 4) shown do not provide any evidence that the rod-like structures shown are SiC nanowires. The morphology is always different, no EDX mapping was performed to confirm
4. Same applies to TEM images (Fig 9). The authors claim that the hexagonal particles indicated by arrows are perpendicularly oriented SiC nanowires. This is simply not true. What I see in these images are flat platelets of regular and irregular morphology, which have nothing to do with the nanowires.
5. The authors have failed to obtain statistically significant data as evidenced by graphs in Fig 2, 5, 8
6. Overall, the message of this paper is very poor, novelty is questionable. This is a technical reports which will not secure any wider interest
Author Response
English is very poor, should be edited professionally.
We have revised the English language throughout the manuscript by a native speaker. The revised places have been marked by blue color in the manuscript.
Optical microscopy images shown do not provide any morphology insights.
We agreed with the reviewer that the optical microscopy images do not provide any morphology insights. However, we just show them to reveal the distribution of the pores in the composites. Due to the large difference between the size of the SiC nanowires (about sub-micron) and pores (about 20 μm), it is difficult to shown them both clearly in the same image. Sorry we did not state it clear in our manuscript. We have revised the manuscript in Line 105-109 as:
“Due to the large difference between the size of the SiC nanowires (about sub-micron) and pores (about 20 μm), it is difficult to shown them both clearly in the same image. In order to shown the effect of the extrusion treatment on the distribution of pores in the composites, optical morphologies of SiCnw/2024Al composite extruded at different temperature have been shown in Fig.3.”
SEM images (Fig 3, 4) shown do not provide any evidence that the rod-like structures shown are SiC nanowires. The morphology is always different, no EDX mapping was performed to confirm
We have added the elements distribution of the Al, Si and C in revised Fig.5. we also added the discussion on the EDX analysis results in Line 136-142 as:
“In order to reveal the distribution of the SiC nanowires in the composites, EDX mapping has been performed, and the distribution of the Al, Si and C elements have been shown in Fig.5. Due to the accuracy of the EDX, the distribution of C element was not very reliable. However, the distribution of Si element was very clear, as shown in Fig.5c. Due to the low content of Si element in 2024Al matrix, the distribution of Si, mainly in morphology of rod-like, could be considered to be equivalent to distribution of SiC nanowires. It also well agreed with our previous research work [15, 17] and literature results [11, 14].
Figure 5. EDX mapping of the polished SiCnw/2024Al composite.(a) SEM image and corresponding distribution of (b) Al, (c) Si and (d) C, respectively.
Same applies to TEM images (Fig 9). The authors claim that the hexagonal particles indicated by arrows are perpendicularly oriented SiC nanowires. This is simply not true. What I see in these images are flat platelets of regular and irregular morphology, which have nothing to do with the nanowires.
Sorry for the confusion. We have revised the manuscript and added more discussion on the TEM results. The presence of the SiC nanowires was further confirmed by the selected area electron diffraction (SAED) patterns, as shown in the revised Fig.12b.
Figure 12. The TEM microstructure of extruded SiCnw/2024Al composite after annealing. (a) Low magnification image, (b) SAED patterns of the SiC nanowire in the dashed circle and (c) SiC nanowires located at the boundaries of Al grains, (d) Few non-directional SiC nanowires.
Moreover, in our previous research work (R. Dong, W. Yang, P. Wu, M. Hussain, Z. Xiu, G. Wu, P. Wang, Microstructure characterization of SiC nanowires as reinforcements in composites, Mater. Charact. 2015, 103, 37–41), the cross-section and axial microstructure of the SiC nanowires embedding into Al matrix have been investigated, and it has been found that the SiC nanowires were composed of a large number of small fragments that are formed by hybrid 3C-SiC and 2H-SiC structures. Therefore, their SAED patterns were composed of two sets of diffraction patterns, corresponding to 2H-SiC and 3C-SiC, respectively, which agreed well with the present observation. The corresponding results have been cited to shown below:
Fig.4 The microstructure characterization of SiC nanowires perpendicular to their axial direction. (a) Cylindrical SiC nanowires; (b) Bamboo shaped SiC nanowires; (c) and (d) Selected area electron diffraction patterns of cylindrical (Fig.4a) and bamboo shaped (Fig.4b) nanowires, respectively. (e) The superposition of the simulated diffraction patterns of 2H-SiC along zone axes (blue spots) and 3C-SiC along [011] zone axes (red spots).
Fig.5 The cross-section microstructure characterization of SiC nanowires. (a) Cylindrical type SiC nanowires; (c) Bamboo shaped SiC nanowires; (b) and (d) Selected area electron diffraction patterns of cylindrical (Fig.5a) and bamboo shaped (Fig.5c) nanowires, respectively. (e) The superposition of the simulated diffraction patterns of 2H-SiC along [0001] zone axes (blue spots) and 3C-SiC along [111] zone axes (red spots).
We have also added the discussion in the manuscript from Line 228 to 233 as:
“In our previous research work [37], the cross-section and axial microstructure of the SiC nanowires embedding into Al matrix have been investigated, and it has been found that the SiC nanowires were composed of a large number of small fragments that are formed by hybrid 3C-SiC and 2H-SiC structures. Therefore, their SAED patterns were composed of two sets of diffraction patterns, corresponding to 2H-SiC and 3C-SiC, respectively, which agreed well with the present observation.”
The authors have failed to obtain statistically significant data as evidenced by graphs in Fig 2, 5, 8
Thanks for the suggestion. We have revised the Fig.4 (former Fig.2) and Fig.14 (former Fig.8) and added the statistically significant data in the figures.
Figure 4. Effect of extrusion temperature on the relative density of the SiCnw/2024Al composite.
Figure 14. The detailed comparison of the yield and tensile strength of the SiCnw/2024Al composite.
Regarding the former Fig.5, it is very sorry that we have not stated clearly about the data of “the average length of SiC nanowires.” We measured the length of the SiC nanowires from SEM images, while the corresponding length would be counted by the Nano Measurer software. After statistical analysis, the normal distribution of the SiC nanowires’ length could be given by the software, as shown in Fig.7b, and the average length of SiC nanowires is already the statistical results. We have added the process in Line 160 to 162 as:
“eventually the length of the SiC nanowires has been counted by the Nano Measurer software. After statistical analysis, the normal distribution of the SiC nanowires’ length could be given by the software, as shown in Fig.7b.”
We also revised the Fig.7 to make it clearer as:
Figure 7. Microstructure of SiCnw/2024Al composite extruded at 560°C after etched by 10% NaOH solution for 480s and corresponding statistical results of the SiC nanowires’ length. (a) Microstructure, (b) Statistical results.
Overall, the message of this paper is very poor, novelty is questionable. This is a technical reports which will not secure any wider interest.
We are sorry that confused information in our previous manuscript misled the reviewer. We have revised the manuscript and added more data and discussion, as stated in above.
Regarding the novelty, SiC nanowires reinforced Al matrix composites have been widely investigated due to the high specific strength and modulus recently. However, the effect of the extrusion temperature on the SiCnw/Al composites, which with high strength and high aspect ratio reinforcement (more than 50), has not been investigated. Our work revealed the effect of the extrusion temperature on the the microstructure evolution and mechanical properties of the SiCnw/2024Al composites, and the corresponding mechanism has been discussed. These results are original and would be helpful for the design of high strength Al matrix composites.
Please see the attachment for the detailed informations.

Reviewer 2 Report
The article falls within the scope of the journal Materials. The paper contains very interesting experimental results. It is of sufficient scientific interest and has originality in its technical content to merit publication. The authors have cited the relevant literature. Methods, interpretations of results and conclusions are correct and novel. In terms of content, the analysis does not raise any objections. The arrangement of work needs improvement. In my opinion, the manuscript is not suitable for publication in its present form.
Detailed comments are provided below.
1 The abstract contains a discussion of research results, which corresponds rather to the discussion of research results and conclusions. The abstract must be improved.
2. The title of the second chapter is not adequate to its content. The chapter also discusses research stands, not just materials and methods.
3. In second chapter, the authors should place photographs or schemes of research stands as well as photos or figures with the dimensions of the samples used in the research.
4. In second chapter, it is not appropriate to send the reader back to [26] reference. Such a reference indicates that the results of the tensile test have already been published and should not be analyzed in this manuscript.
5. How was the density and porosity investigated in the manuscript? This is not sufficiently explained.
6. The third chapter of the Results and discussion should be divided into two subchapters. In the first one should be presented an analysis of the impact of extrusion temperature on the microstructure, in the second - on mechanical behavior.
7. In the conclusions, it would be desirable to specify conclusions pointwise rather than presenting them in a descriptive form.
8, The manuscript should be checked by a native speaker.
Author Response
The abstract contains a discussion of research results, which corresponds rather to the discussion of research results and conclusions. The abstract must be improved.
We have revised the abstract according to the reviewer’s suggestion in Line 13 to Line 20 as:
“In the present research work, effect of extrusion temperature from 480 to 560ºC on the microstructure and mechanical behavior of the SiCnw/2024Al composite (15vol.%) has been explored. It has been found that extrusion at higher temperature (above 520ºC) was beneficial for the densification of the composite, while the residual average length and alignment of the SiC nanowires were also increased with the extrusion temperature. Moreover, higher extrusion temperature was helpful for the mechanical strength of the SiCnw/2024Al composite, and the peak-aged SiCnw/2024Al composite extruded at 560ºC revealed highest strength (709.4 MPa) and elastic modulus (109.8 GPa).”
The title of the second chapter is not adequate to its content. The chapter also discusses research stands, not just materials and methods.
We have revised the title of the second chapter from “materials and methods” to be “2. Experimental design, materials and methods” according to the reviewer’s suggestion. We also added sub-headings of “2.1 Materials and Composite fabrication process”, “2.2 Microstructure Characterization” and ” 2.3 Mechanical properties testing. ”
In second chapter, the authors should place photographs or schemes of research stands as well as photos or figures with the dimensions of the samples used in the research.
We have added the schemes of research stands in Fig.1 as:
Figure 1. The schematic diagram of preparation and extrusion process of SiCnw/2024Al composite.
We also added the dimensions of the density (10×10×2mm, Line 87), elastic modulus (3×4×36mm, Line 95) and tensile test samples (Line 98 and Fig.2) used in the research.
Figure 2. Dimensions of tensile test samples (2mm in thickness).
In second chapter, it is not appropriate to send the reader back to [26] reference. Such a reference indicates that the results of the tensile test have already been published and should not be analyzed in this manuscript.
Thanks for the suggestion. The reference [26] was just to show the dimensions of the tensile test samples, not the tensile test results. We have revised the manuscript to make it clearer in Line 96 to 100 as:
“Paralleling to the extrusion direction, tensile properties were measured on universal electrical tensile testing machine (Instron 5569, Instron Co., USA) with a speed of 0.5 mm/min, and eventually the SEM fracture surface was observed by Quanta 200. The tensile samples (2mm in thickness) were prepared from the composites, as shown in Fig.2. Six samples extruded at different temperature were tested for each set of composite samples to improve the statistical significance of the results.”
How was the density and porosity investigated in the manuscript? This is not sufficiently explained.
We measured the density firstly by the Archimedes principle, and then the densities of the composites before and after extrusion were obtained. According to the microstructure observation the sample extruded at 560°C was rather dense without presence of pores. Moreover, density test result indicated that the samples extrude at 520 and 560°C was stable, indicating no improvement in densification effect. Therefore, the density of the samples extruded at 560°C was considered as the “theoretical” density. By comparison of the measured true density and the “theoretical” density, the relative density was obtained.
We have revised the manuscript in Line 86 to 90 and Line 122 to 129 as:
“The density of all samples (10×10×2mm) was measured using Archimedes principle, and four samples have been tested to improve the statistical significance of the results. The relative density of the composites before and after extrusion was calculated by comparison of the measured true density and the “theoretical” density of the samples extruded at 560°C.”
“After extrusion, the measured density of the composites was initially increased with and eventually stabilized above 520 °C with the increase of the extrusion temperature, indicating that the composites have reached full densification after extrusion at 560°C. Therefore, the measured true density of the composites in various states were divided to the “theoretical” density of the samples extruded at 560°C, and then the relative density has been obtained, as shown in Fig.4. It is clear that the composites have been densified after the extrusion, and the densification effect was also improved with the extrusion temperature, which agreed well with optical microstructure observation (Fig.2).”
The third chapter of the Results and discussion should be divided into two subchapters. In the first one should be presented an analysis of the impact of extrusion temperature on the microstructure, in the second - on mechanical behavior.
We have divided the Results and discussion to two subchapters, “3.1 Effect of Extrusion temperature on the Microstructure of SiCnw/2024Al Composite ” (Line 104) and “3.2 Effect of Extrusion temperature on the Mechanical behavior of SiCnw/2024Al Composite” (line 244).
In the conclusions, it would be desirable to specify conclusions pointwise rather than presenting them in a descriptive form.
We have revised the conclusion to pointwise format according to reviewer’s comment as:
“In the present work, the effect of extrusion temperature from 480 to 560ºC on the microstructure and mechanical behavior of the 15 vol.% SiCnw/2024Al composite prepared by pressure infiltration methods have been investigated, and following conclusions have been obtained:
After extrusion treatment, the porosity of the SiCnw/2024Al composite has been significantly decreased, and the composites have reached full densification after extrusion at 560°C. Moreover, the average length of the residual SiC nanowires was also increased with the extrusion temperature. the distribution of the SiC nanowires has been changed from random to alignment along the extrusion direction, while the alignment degree of the SiC nanowires was increased with the extrusion temperature according to XRD analysis. TEM observation indicated that the presence of the SiC nanowires was mainly located at the boundary of Al grains due to their “non-deformable” character. Regardless of the heat treatment, the yield and tensile strengths of the SiCnw/2024Al composite were increased with the extrusion temperature, and many dimples and tearing ridges were observed in the fracture surface. The peak-aged SiCnw/2024Al composite extruded at 560ºC revealed highest strength (709.4 MPa) and elastic modulus (109.8 GPa). Based on the modified shear-lag mode, the effect of the extrusion temperature on the mechanical properties of the SiCnw/2024Al composite has been discussed, and the improved strength was mainly due to the better alignment and longer average length of the SiC nanowires at higher extrusion temperature. ”
The manuscript should be checked by a native speaker.
We have revised the English language throughout the manuscript by a native speaker. The revised places have been marked by blue color in the manuscript.
Please see the attachment

Reviewer 3 Report
The paper entitled ‘Effect of extrusion temperature on the microstructure and mechanical behavior of SiCnw/2024Al composites’ is suitable for the Materials journal. The manuscript is exhaustively written and the results are supported by well prepared and discussed experimental data. In this paper, 15 vol.% SiCnw/2024Al composites have been hot-extruded in the temperature from 480 to 560oC, and the effect of the extrusion temperature on the microstructure and mechanical behaviour of the SiCnw/2024Al composites have been discussed. As authors underlined, the peak-aged SiCnw/2024Al composite demonstrated both high strength and elastic modulus, which made it very attractive for aerospace application.
Author Response
We have revised the English language throughout the manuscript by a native speaker. The revised places have been marked by blue color in the manuscript.
Round 2
Reviewer 1 Report
The revised paper is obviously better, though there are still other issues to be dealt with.
- I would like to see TEM and SEM images of ra SiC nanowires, because what the authors show in Fig 12 a and d does not really correspond with the SEM images of the wires.
- Charts should demonstrate error value
- The literature review is very limited. Some recent paper on similar materials are suggested for citation:
- Mishra et al, ACS Applied Nano Materials 2018, 1, 7, 3715-3723
- Foisal et al, RSC Adv., 2018,8, 15310-15314
- Nguyen et al, RSC Adv., 2018,8, 3009-3013
- Hart et al, ACS Appl. Mater. Interfaces201791513742-13750
Author Response
I would like to see TEM and SEM images of raw SiC nanowires, because what the authors show in Fig 12 a and d does not really correspond with the SEM images of the wires.We have added the SEM and TEM images of the raw SiC nanowires in the revised manuscript as Fig.1:
Figure 1. Microstructure of the raw SiC nanowires ((a) SEM image and (b) TEM image) used in the present research work
We also add the description of the SiC nanowires in Line 70 to 71 as:
“The distribution range of length and diameter of the SiC nanowires were from 10 to 50 μm (Fig.1a) and 100 to 500 nm (Fig.1b), respectively.”
The diameter and length of the SiC nanowires were not uniform, which may affect the view results. Moreover, the SEM and TEM of the SiC nanowires were observed in a certain section, which may affect the view results.
- Charts should demonstrate error value
We also add the error values according to the review’s suggestion in Fig.4, Fig.12 and Fig.15.
Figure 4. Effect of extrusion temperature on the relative density of the SiCnw/2024Al composite.
Figure 12. The (111)/(220) PVR of the SiC nanowires in the SiCnw/2024Al composite perpendicular and parallel to the extrusion direction.
Figure 15. The detailed comparison of the yield and tensile strength of the SiCnw/2024Al composite.
The literature review is very limited. Some recent paper on similar materials are suggested for citation:
(1) Mishra et al, ACS Applied Nano Materials 2018, 1, 7, 3715-3723
(2) Foisal et al, RSC Adv., 2018,8, 15310-15314
(3) Nguyen et al, RSC Adv., 2018,8, 3009-3013
(4) Hart et al, ACS Appl. Mater. Interfaces201791513742-13750
Thanks for the suggestion. We have cited the above four important work in Line 26 to 28 as:
“As another important one-dimensional materials, application of the SiC nanowires in functional sensors or MEMS have been widely investigated [7-10].”
R. Mishra, J.B. Tracy. Sequential Actuation of Shape-Memory Polymers through Wavelength-Selective Photothermal Heating of Gold Nanospheres and Nanorods. ACS Appl. Nano Mater. 2018, 17, 3063-3067. R.M. Foisal, H-P. Phan, T. Dinh, T-K. Nguyen, N-T. Nguyen, D.V. Dao. A rapid and cost-effective metallization technique for 3C–SiC MEMS using direct wire bonding. RSC Adv. 2018, 8, 15310-15314. T-K. Nguyen, H-P. Phan, J. Han, T. Dinh, A.R.M. Foisal, S. Dimitrijev, Y. Zhu, N-T. Nguyen, D.V. Dao. Highly sensitive p-type 4H-SiC van der Pauw sensor. RSC Adv. 2018, 8, 3009-3013. H.C. Hart, R. Koizumi, J. Hamel, P.S. Owuor, Y. Ito, S. Ozden, S. Bhowmick, S.A.S. Amanulla, T. Tsafack, K. Keyshar, R. Mital, J. Hurst, R. Vajtai, C.S.Tiwary, P.M. Ajayan. Velcro-Inspired SiC Fuzzy Fibers for Aerospace Applications. ACS Appl. Mater. Interfaces. 2017, 9, 15, 13742-13750.Moderate English changes required
We have revised the English language throughout the manuscript by a native speaker again, and the revised places have been marked by blue color in the manuscript.
Please see the attachment for detailed information.
